

# Influence of acute combined physical and cognitive exercise on cognitive function: an NIRS study

Zhiguang Ji[1,2], Tian Feng[3], Lingnan Mei[4], Anmin Li[1] and Chunhua Zhang[1]

[1] School of Kinesiology, Shanghai University of Sport, Shanghai, China
[2] Shanghai University of Medicine & Health Sciences, Shanghai, China
[3] Physical Education College of Zhengzhou University, Zhengzhou, China
[4] School of Physical Education, Central China Normal University, Wuhan, China

## ABSTRACT

The purpose of this study is to investigate the effects of different types of acute exercise on cognitive function and cerebral oxygenation. A within-subject design was adopted. In total, 20 healthy older adults were enrolled in the study. They came to the laboratory individually on four separate days and completed four conditions of activity. Four conditions were sedentary reading control (RC), cognitive exercise (CE), physical exercise (PE) and cognitive + physical exercise (CE + PE). During these visits, participants completed the Stroop task before and immediately after the experimental condition, which consisted of 15 min of aerobic exercise, verbal fluency task (VFT), and dual task. The Stroop task included the following two conditions: a naming condition and an executive condition. The fNIRS is an optical method using near-infrared light to measure relative changes of oxygenated ($O_2$Hb) and deoxygenated (HHb) hemoglobin in the cortex. The results indicate that acute exercise facilitates performance for executive tasks, not only combined cognition, but also the different results between combined exercise and single exercise. The fNIRS findings showed that acute single exercise influences oxygenation for executive tasks but not for naming tasks. Greater improvement was observed in the post-exercise session of combined exercise during the modified Stroop. These findings demonstrate that acute single exercise, single cognition exercise, and combined exercise enhanced the performance of the inhibition control task. Only acute combined exercise has a general facilitative effect on inhibition control. Combined exercise was shown to be superior to single exercise for task-efficient cerebral oxygenation and improved oxygen utilization during cortical activation in older individuals. Also, to maximize the performance of cognition it may be important for older adults to take part in more cognitive demand exercise or take more kinds of exercise.

# INTRODUCTION

Some meta-analytic reviews and interventions have demonstrated that acute exercises including short duration exercises have positive effects on cognition in older adults (*Chang et al., 2012*; *Peiffer et al., 2015*). However, the mechanism by which acute exercise influences

Corresponding authors
Zhiguang Ji, ji_zg09@126.com
Chunhua Zhang,
zch20080808@126.com

cognitive function is unclear. Inhibition refers to the process of controlling one's attention, behavior, thoughts, and/or emotions to override a strong internal predisposition or external lure (*Barkley, 2012*). *Bediz et al. (2016)* proposed that the change of the prefrontal cortex (PFC) oxygenation may be one of the mechanisms. In recent studies that take advantage of human brain imaging, a change in brain oxygenation has been demonstrated in pre-and post-invention tests. Most of those studies showed an increase in oxygenation of PFC following exercise (*Jung et al., 2015*). An increase in prefrontal area Oxy-Hb response tends to correlate with better performance in cognitive test (*Coetsee, 2015*; *Endo et al., 2013*). Near infrared spectroscopy (NIRS) is a non-invasive method for measuring cerebral hemodynamic response to cognitive tasks. More and more studies employed NIRS to record the brain oxygenation during or post intervention conditions. Recent findings from fNIRS studies showed that an acute bout of exercise, regardless of intensity, improves performance on executive function and may induce different prefrontal area oxygenation. *Li et al. (2005)* reported a higher dorsolateral PFC activation during working memory task by using fNIRS. *Endo et al. (2013)* demonstrated that exercise at moderate intensity for 15 min may improve cognitive function and neural activity in the prefrontal areas. *Byun et al. (2014)* showed the acute bout of mild exercise can improve Stroop performance, which evoked cortical activations on the left dorsolateral prefrontal cortex and frontopolar area.

Despite accumulating evidence for the positive effects of acute exercise on cognition, most of the results were derived from studies focusing on aerobic exercise. *Pesce (2012)* claimed that future research should not only target quantitative exercise characteristics, but also emphasize the qualitative aspects of exercise, such as the cognitive features during exercise.

Moreover, there are limited number of studies investigating the effects of different types of exercise. *Hsieh et al. (2016)* found that acute resistance exercise enhances attention control and working memory in both young and older males. Lately, researchers have demonstrated that exercise with more cognitive demands in term of exercise mode was more effective for completing cognitive task (*Budde et al., 2008*; *Chen et al., 2017*; *Eggenberger et al., 2015*). Some studies on chronic exercise have proposed that a combined cognitive and physical exercise may result in greater oxygenation of PFC than either intervention alone. Our earlier work showed that a high cognitive demand may be a potential factor that influences the efficacy of exercise on cognitive function (*Ji et al., 2017*). However, whether a combination of cognitive and physical exercise exceeds single exercises in acute exercise remains unsolved. To date, previous studies showed unclear results. *Holtzer et al. (2011)* provided the evidence that Oxy-Hb levels are increased in the PFC during Walk-While-Talk compared with Normal Walk in young and older individuals. Later studies are also consistent with the results that greater brain activation is observed following the combined cognitive and physical exercise (*Holtzer et al., 2015*; *Holtzer et al., 2017*; *Mirelman et al., 2014*). In addition, few studies compared the cognitive function after acute exercises with different cognitive demands. According to neurovascular coupling theory (NVC), the metabolic activity can be assessed as an increase in cerebral blood flow (*Attwell et al., 2010*). We assume that the cognitive characteristic of an exercise might be

responsible for the significant differences. The first aim of this study is to reveal the effect of the different exercises on the oxygenation and cognitive function.

While many aspects of cognitive function are associated with aging, the cognitive domain of executive function is particularly relevant. *Yanagisawa et al. (2010)* revealed a selective improvement on the executive task of the Stroop task following acute exercise, which may depend on heightened activation of dorsolateral prefrontal cortex. Some meta-analyses indicate that acute exercise can be used to promote performance in subsequent higher-order cognition such as attention, executive control, and short-term memory (*Chang et al., 2012*; *Hung et al., 2013*; *Pesce & Audiffren, 2011*). However, *Chang et al. (2017)* suggested that the improvement in cognitive function is generalized rather than selective improvement after moderate intensity acute exercise. In our earlier cross-sectional research, we also found different benefit for different process level of cognition between Tai Chi and brisk walking (*Ji et al., 2017*). In this study, the Tai chi group showed better performance of the naming and the executive conditions than the brisk walking group. These findings demonstrated that regular participation in brisk walking and Tai Chi have significant beneficial effects on executive function and fitness. A meta-analysis by *Ludyga et al. (2016)* revealed that preadolescent children and older adults who participate in a single moderately-intense aerobic exercise may get less benefit than younger adults, while the aerobic activities have potential neuroenhancing effect as higher education, professional training and work temporarily place high demands on the executive function system.

The aim of this study is to address the gap in knowledge regarding the effects of acute combined and single exercise on cognitive function and cerebral oxygenation. A major question is to explore the influence of acute combined physical and cognitive exercise on cognitive function. We hypothesize that combined exercise would lead to a general pre-activation of cognitive-related prefrontal cortex (PFC) oxygenation and would be more effective in improving the speed and accuracy of the modified Stroop compared to single exercise.

## MATERIALS AND METHODS

### Ethical approval
The study was conducted ethically and received approval from the Ethics Committee of Shanghai University of Sport (No. 2017032).

### Participants
Twenty participants (nine female, mean age = 67 ± 3.2 year) were recruited from the Shanghai Yanji community via flyers and personal referrals. All participants provided written informed consent that is obtained by the Shanghai University of Sport. Additionally, participants finished a health history questionnaire, reported being free from any neurological diseases, and had normal or corrected-to-normal vision, normal color vision. Table 1 presents participant demographics.

### Experimental procedure
The participants joined the laboratory individually on four separate days, the interval between each visit is 3-day. On the first visit, participants completed an informed consent

**Table 1 Participants' characteristics.**

|  | MEAN | SD |
|---|---|---|
| Age, years | 65.60 | 1.32 |
| Weight, kg | 59.35 | 6.15 |
| Hight, cm | 161.75 | 5.58 |
| BMI | 22.66 | 1.71 |
| MMSN | 27.90 | 0.62 |
| Education, years | 11.25 | 1.48 |
| M/F | 0.45 | / |
| VO2peak | 26.00 | 2.09 |

**Notes.**
BMI, body mass index; MMSE, Mini-Mental State Exam; IPAQ, International Physical Activity Questionnaire.

and the Physical Activity Readiness Questionnaire (PAR-Q) to confirmed the health issues of the participants that may be exacerbated by acute exercise (*Pontifex et al., 2009*). Participants were instructed to completed a cardiorespiratory fitness test that was used by the YMCA cycle ergometry protocol (*USA Yot, 2000*) and cognitive function was assessed using the Mini-mental State Examination (*Tombaugh & Mcintyre, 1992*).

On laboratory visits 1–4, participants completed a reading control condition (RC) and three treatment conditions, cognitive exercise (CE), physical exercise (PE) and cognitive+physical exercise (CE + PE). To minimize any order or learning effects, the condition order was counterbalanced across participants. The reading control condition was complicated read materials related to exercise. Participants completed the Stroop task before and immediately after the control and treatment conditions (when the heart rate returned to rest level). During the Stroop task, oxygenation changes in the cortex were measured throughout the session using fNIRS. Additionally, we make sure that most anterior and ventral pairs were positioned in similar locations for each participant and in each visit. Before the experimental trials, participants completed practice trials to become familiarized and an 85% correct rate was allowed. Next, participants were instructed to complete an acute session, which included 5-min warm-up to become familiarized with the test procedure. For the RC, participants sit quietly in a chair and completed Verbal Fluency Task (VFT) for approximately 15 min. Participants were asked to think of as many animal/sport/country/vegetable/fruit names as possible in each 2 min with 1 min interval. The single exercise procedure consisted of three stages: a 5-min warm-up, a 15-min primary exercising (65% heart rate returned), and a 5-min cool-down. The CE + PE condition which combined walking at the same velocity as the 15-min primary exercising of PE condition with the VFT. The control condition was proposed to keep an arousal level compared to the exercise condition.

## Cardiorespiratory fitness assessment

The YMCA protocol involves three consecutive stages. The first level was at a power and pedaling rate of 25W and 50 rpm for 3 min (150 kpm/min). The heart rate (HR) was assessed using a Polar heart rate monitor (Sport Tester PE 3000; Polar Electro Oy, Kempele, Finland). The next two exercise stages were decided according to the last minute of cycling.

Two heart rate values were recorded during the final 15–30 s of the first and second stages. These two heart rate values, along with the YMCA equations, the individual's body weight, and age, were used to calculate the estimated $VO_2$ peak (*Beekley et al., 2004*).

## Cognitive task

The computerized modified Stroop task was based on the Stroop task and included the following two tasks: a naming task (non-executive) and an executive task (*Bohnen, Twijnstra & Jolles, 1992*; *Dupuy et al., 2015*). Each block lasted 60s and was interspersed with 60-s resting blocks. The blocks were always offered in the same order: Rest–Naming–Rest–Executive–Rest–Naming–Rest–Executive. All trials began with a fixation cross that appeared at the center of the computer screen for 1,000 ms, and visual stimuli for a duration of 2,000 ms.

Participants provided their responses by the response key mapping that was used in the task (i.e., red—J key, blue—F key). In the naming task, a visual stimulus was presented as ink colors of XXX and participants were instructed to indicate the color of the ink by pressing a response key (i.e., red—J key, blue—F key). In the executive task, part of the presented stimuli is the ink colors of color names, such as ''RED'' printed in blue ink, and the other part of the stimuli were words surrounded by rectangles, the participants were instructed to read the word, ignore the color of the ink (Fig. 1). Participants did not speak during either the naming or the executive task to avoid artefacts resulting from speaking. Reaction times (in milliseconds) and accuracy values were recorded for all tasks. Dependent variables were reaction times and the number of errors committed (%).

## fNIRS recordings

A multi-channel continuous wave fNIRS instrument (NIRScout, NIRx Medical Technologies LLC, USA) was used. The system collects (NIRStar 15.2) data that was sampled at 3.91 Hz. The fNIRS sensor had 20 channels created by 8 LED light sources and 8 photodetectors, the source-detector separation was 3 cm. The probe locations and specific brain regions were selected according to studies of *Okamoto et al. (2004)* and *Tsuzuki et al. (2007)*. NIRS-SPM (*Ye et al., 2009*) and the software which integrates a 3D digitizer (FASTRAK-Polhemus, Polhemus, VT, USA) was used to complete the 3-dimensional spatial registration of NIRS channel locations.

Four different approximate regions of interests (ROI) were defined, but do not refer exactly to the underlying brain regions. Areas of interest were the left ventrolateral prefrontal cortex (VLPFC) (channels 1–4), the left dorsolateral prefrontal cortex (DLPFC) (channels 5, 6, 7 and 11), the right DLPFC (channels 13–16) and the posterior right VLPFC (channels 17–20) for both hemispheres (Fig. 2). The fNIRS transmitters were tightly secured with a black bandage wrapped to ensure that there was no interference of extraneous light and to limit movement during cognitive task. Variables of interest were relative changes in concentration of Oxy-Hb and Deoxy-Hb compared to the baseline (1 min at rest before the computerized Stroop task) (*Gagnon et al., 2012*; *Laguë-Beauvais et al., 2013*).

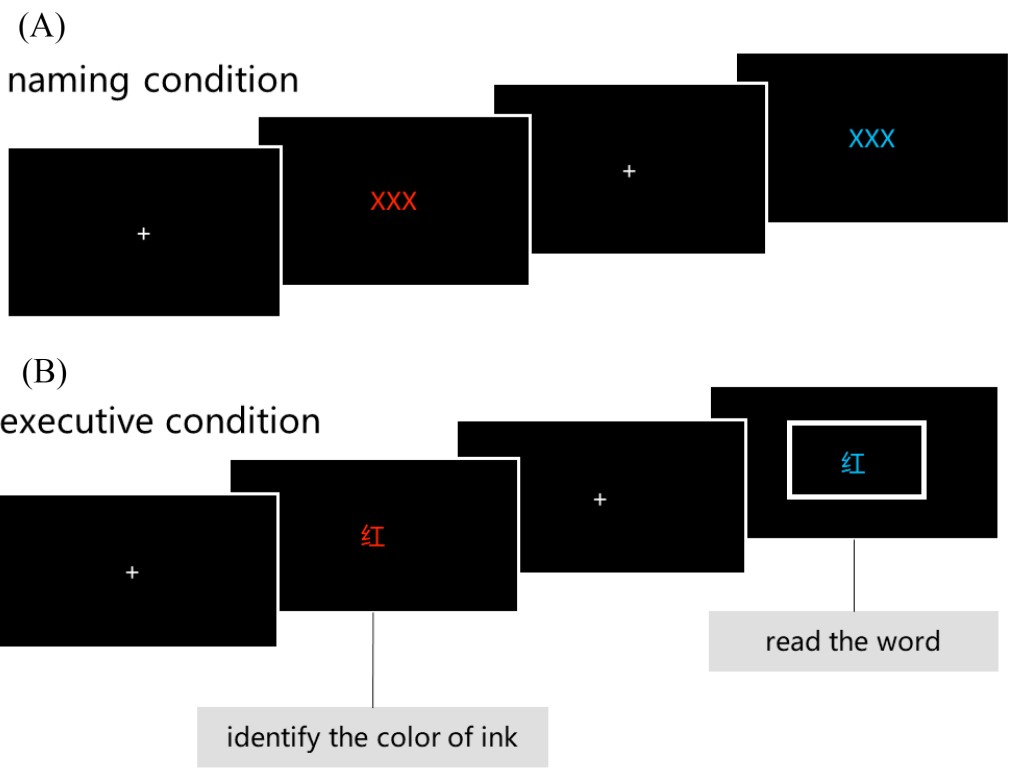

**Figure 1  Illustration of the modified Stroop task.** (A) The naming condition of the Modified Stroop task, (B) the executive conditions of the Modified Stroop task.

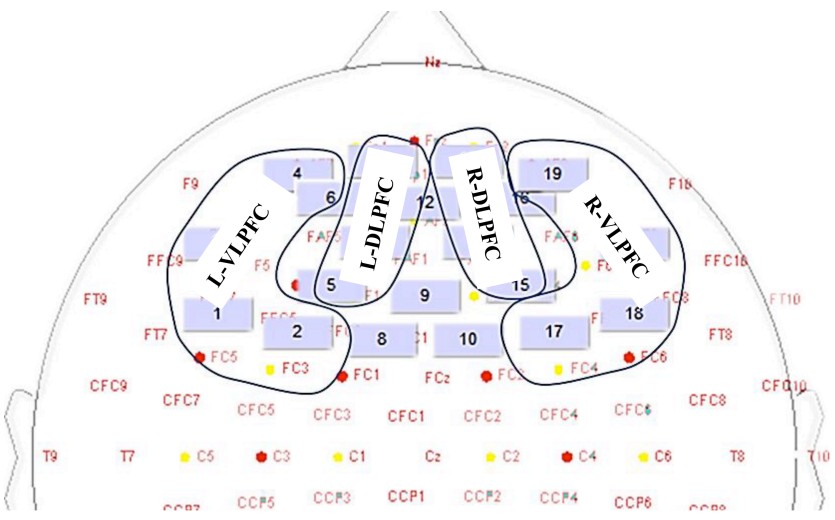

**Figure 2  The location of the fNIRS probes and the ROI.**

## Analysis

### fNIRS analysis

NIRS data analysis was performed in nirsLAB based on SPM and NIRS_SPM with additional modules for ANOVAs. The fNIRS signals were bandpass filtered between 0.01 Hz and 0.2 Hz to remove baseline drift, artefacts and physiological noise. The analysis of fNIRS signal was performed by block (i.e., by task, including all trials), without separating out error trials. For the fNIRS data, we calculated the value of the Oxy-Hb change, by subtracting the mean Oxy-Hb of the pre-intervention task period from the mean Oxy-Hb of the post-intervention task period. Previous studies revealed that the changes in Oxy-Hb are more sensitive to changes in regional CBF than Deoxy-Hb, while the changes in Deoxy-Hb are mainly determined by venous oxygenation and blood volume than blood flow changes (*Hoshi, Kobayashi & Tamura, 2001*). Thus, we only present the results of Oxy-Hb in the current study.

### Statistical analysis

Behavioral and Oxy-Hb data were analyzed in SPSS (v19.0 ) using a 3 (conditions) × 2 (assessments) repeated-measures ANOVA. ANOVAs were performed on each task (Naming and Executive) and post-hoc test P values were Bonferroni corrected for multiple comparisons.

## RESULTS

### Behavioral findings

When the response time was considered, in the naming task, there was a significant main effects of time ($F_{(1,76)} = 25.49$, $p < 0.001$, $\eta^2 = 0.25$) and a interaction between time and condition ($F_{(3,76)} = 4.31$, $p < 0.001$, $\eta^2 = 0.15$). No significant main effects of condition ($F_{(3,76)} = 0.63$, $p = 0.621$). The post-hoc comparison showed that the participants reduced reaction time from pre-test to post-test ($p = 0.043$) in CE + PE condition. In the executive task, there was a significant main effects of time ($F_{(1,76)} = 39.56$, $p < 0.001$, $\eta^2 = 0.34$) and a interaction between time and condition ($F_{(3,76)} = 11.58$, $p < 0.001$, $\eta^2 = 0.31$). No main effect of condition ($F_{(3,76)} = 1.25$, $p = 0.303$) was observed. The post-hoc comparison showed that the participants reduced reaction time from pre-test to post-test in CE + PE ($p = 0.031$) and PE ($p = 0.043$) condition. All results are shown in Fig. 3.

When the accuracy was considered, in the naming task, there was no significant main effects of time ($F_{(1,76)} = 0.04$, $p = 0.852$), condition ($F_{(3,76)} = 0.57$, $p = 0.648$) and interaction between time and condition ($F_{(3,76)} = 0.35$, $p = 0.798$). In the executive task, there was no significant main effects of time ($F_{(1,76)} = 0.28$, $p = 0.601$), condition ($F_{(3,76)} = 0.67$, $p = 0.581$) and interaction between time and condition ($F_{(3,76)} = 0.19$, $p = 0.915$).

### fNIRS findings

Figure 4 presents the effects of four conditions on task-related changes in Oxy-Hb levels. According to these analyses, in the naming task, there was a significant main effects of condition ($F_{(3,76)} = 10.45$, $p < 0.001$, $\eta^2 = 0.29$) and an interaction between ROI and

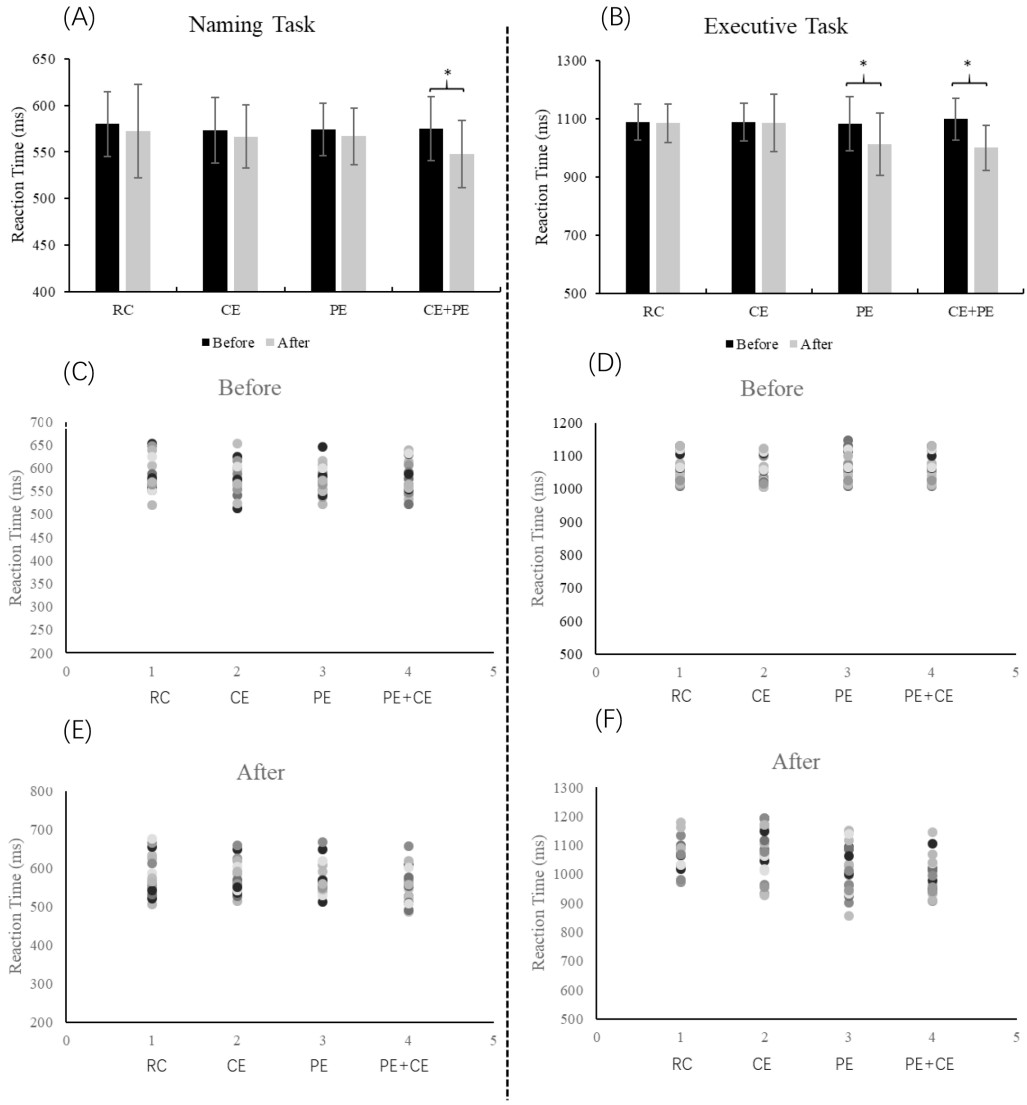

**Figure 3** **Reaction time of the naming task and executive task in the modified Stroop task.** RC, Reading control condition; CE, cognitive exercise; PE, Physical exercise; CE + PE, cognitive + physical exercise. Reaction time of the naming task (A) and executive task (B); (C) scatter plot of the pre-test during naming task; (D) scatter plot of the pre-test during executive task; (E) scatter plot of the post-test during naming task; (F) scatter plot that the post-test during executive task. *Significant difference between pro-test and post-test, $p < 0.05$.

condition ($F_{(9,228)} = 2.13$, $p = 0.028$, $\eta^2 = 0.08$). No significant main effects of ROI were observed ($F_{(3,228)} = 1.80$, $p = 0.153$). Post-hoc comparison showed, in the L-VLPFC area, CE conditions were significantly higher than RC conditions ($p < 0.001$). In the R-VLPFC area, CE + PE conditions were significantly higher than RC conditions ($p < 0.001$), CE + PE conditions were significantly higher than CE ($p = 0.033$), and PE were significantly higher than RC ($p = 0.041$).
(A)

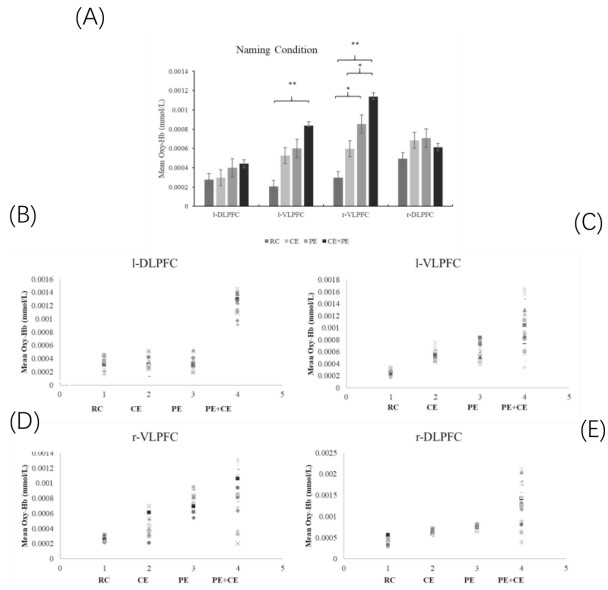

(B)

(C)

(D)

(E)

(F)

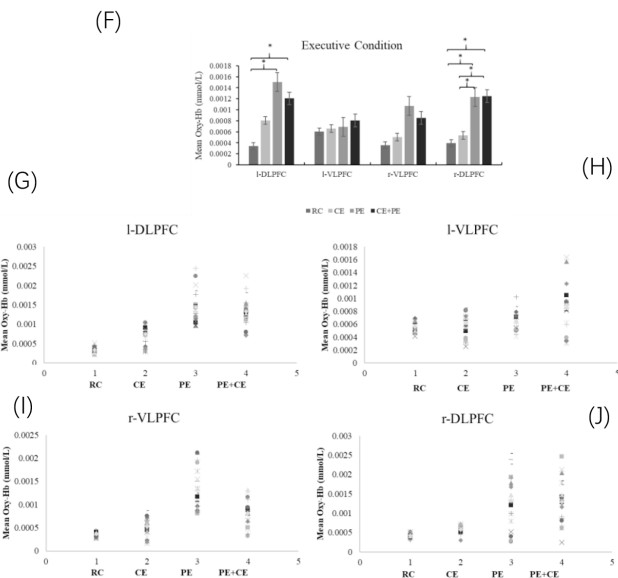

(G)

(H)

(I)

(J)

**Figure 4 The interaction effects of the naming task and executive task on task-related changes in Oxy-Hb levels.** RC, Reading control condition; CE, cognitive exercise; PE, Physical exercise; CE + PE, cognitive + physical exercise. Oxy-Hb levels of the naming task (A) and executive task (F); (B) scatter plot of the Oxy-Hb levels of l-DLPFC during the naming task; (C) scatter plot of the Oxy-Hb levels of l-VLPF during naming task; (D) scatter plot of the Oxy-Hb levels of r-VLPF during the naming task; (G) scatter plot of the Oxy-Hb levels of r-DLPF during the executive task; (H) scatter plot of the Oxy-Hb levels of l-DLPFC during executive task; (I) scatter plot of the Oxy-Hb levels of l-VLPF during executive task; (J) scatter plot of the Oxy-Hb levels of r-DLPF during executive task. *Significant difference between four conditions, $p < 0.05$.

In executive task, there was a significant main effect of condition ($F_{(3,76)} = 7.68$, $p < 0.001$, $\eta^2 = 0.23$), ROI ($F_{(3,228)} = 3.08$, $p = 0.028$, $\eta^2 = 0.04$) and an interaction between ROI and condition ($F_{(9,228)} = 1.97$, $p = 0.043$, $\eta^2 = 0.07$). Post-hoc test revealed that, in L-DLPFC, PE ($p = 0.031$) and CE + PE ($p = 0.035$) showed greater oxygenation levels than RC. In R-DLPFC, PE showed greater level than RC ($p = 0.031$) and CE ($p = 0.042$), and CE + PE showed greater than level RC ($p = 0.041$) and CE ($p = 0.043$). No other significant difference was observed between two intervention conditions. All results of fNIRS are shown in Fig. 4.

## DISCUSSION

The purpose of this study is to determine whether different types of acute exercise result in a general or selective improvement in cognitive function and to investigate potential neurobiological mechanisms by examining cognitive-related prefrontal cortex oxygenation. The findings reveal that acute exercise facilitates performance for executive tasks, not only combined cognition, but also the different results between combined exercise and single exercise. On one hand, single exercise showed selective effects for task requiring executive function (i.e., executive task). Additionally, the fNIRS findings demonstrate that acute single exercise influences oxygenation for executive tasks, but does not influence oxygenation for naming tasks. On the other hand, a general improvement hypothesis was supported by the results of acute combined exercise. In this study, higher oxy-Hb levels and greater increments in the post-exercise session of combined exercise during the modified Stroop suggested a better performance of the naming and executive blocks in Stroop task. In contrast, in the control and cognitive condition, no significant improvement in inhibition control and oxy-Hb levels were observed between pre-test and post-test.

Behavioral measurements showed that the older adults demonstrated a consistently better performance for executive task after both combined and single exercise compared with the control and cognition condition. The effect of acute single exercise on executive function has been demonstrated in previous studies. *Browne et al. (2016)* and *Peruyero et al. (2017)* found acute aerobic exercise improved the inhibitory control in adolescents. The improvement also has be found in study among older adults. *Ludyga et al. (2016)* provided evidence that older adults can improve performance of the task demanding high executive control by an acute aerobic exercise. Additionally, the obtained effect sizes were not influenced by factors include age, aerobic fitness and the executive function component. Results of acute aerobic exercise by *Peiffer et al. (2015)* indicated that older adults improved performance in a modified flanker task after both moderate and vigorous intensity aerobic exercise and these effects would last 30 min post-exercise. Recent studies suggested that other exercises may also improve executive function in older adults. Acute resistance exercise benefited working memory measured by Sternberg working memory in older males (*Hsieh et al., 2016*). The results supported a selective facilitation rather than a general one (*Hsieh et al., 2016*). However, single exercises were used in their intervention studies. The effects of acute exercises combined with cognitive training on cognitive function was unclear. Interestingly, we find that acute combined exercise enhances

cognitive performance in both naming task and executive task, further supporting a general improvement hypothesis of acute exercise rather than a selective improvement in cognition. Altered prefrontal cortex oxygenation might be a possible underlying mechanism of the beneficial Stroop effect associated with acute exercise. In contrast, no significant improvement was observed in control and cognitive condition. These results were similar to previous studies which showed that a control group that reads for the same length of time has no facilitative effects on the executive functions (*Chang et al., 2011*).

In this study, we observed significant oxy-Hb signal increases associated with the execution (naming contrast) in the LPFC covering the bilateral VLPFC and DLPFC. This spatially specific activation pattern is consistent with the results of previous fNIRS studies on Stroop interference, reporting bilateral LPFC activity (*Schroeter et al., 2004*; *Yanagisawa et al., 2010*). Previous fMRI studies have also consistently showed LPFC activation reflecting Stroop interference along with activation of ACC (*Derrfuss et al., 2005*). In the related literature, it has been demonstrated that the oxy-Hb level increases in PFC during post-exercise cognitive task. A previous task-based fMRI study (*Newman et al., 2003*) showed that although both the right and left PFC were activated by a planning task, the activation in the right DLPFC was sensitive to task difficulty. Moreover, *Yanagisawa et al. (2010)* concluded that the LPFC activation may reflect interference processing/response inhibition, and suggested that the left LPFC is likely the neural substrate for the improved Stroop performance. In our study, the oxy-Hb level increases also in LPFC, the brain areas of Stroop-interference-related activation were not only left LPFC but also right LPFC. A potential explanation for the shift of the action in this finding might be due to "reduced lateralization" in old age (*Cabeza, 2002*). *Cabeza et al. (2002)* showed that the bilateral activation during tasks was observed in old people, while in young people the two areas are clearly defined and have separate roles.

fNIRS is a reliable method to explore the mechanisms of physical exercise on brain oxygenation in relation to cognitive performance for decades. However, studies demonstrating the difference between combined exercise and single exercise are very limited and these mechanisms are not well understood. The behavioral data of this study demonstrated that acute combined exercise has greater benefits on naming task. The results of fNIRS showed that combined exercise could cause higher oxygenation responses for naming task in VLPFC. The VLPFC is the end area of the ventral pathway involved in processing/recognition of visual stimuli, while the DLPFC is implicated in the choice, manipulation, and monitoring of information (*Itakura et al., 2017*). *Milham, Banich & Barad (2003)* reported that the DLPFC were more active for executive function than VLPFC. The difference of VLPFC may be the potential cause of different performance in naming tasks.

This study demonstrates the effects of different acute exercise on cognitive function, but some potential limitations should be considered when interpreting our findings. Firstly, additional subcomponents of executive function should be investigated, including working memory, updating, switching, and planning, and their responses to exercise may be specific. Secondly, the number of the subjects in the fNIRS experiments was small. Although the within-subject design suffers less from variation, a larger sample size would be more

powerful. Finally, artefacts of physical activity can be removed by filtering. Artefacts due to speaking may have impacted the results, but the language area was not our ROI. Further studies could optimize task modules to minimize the impact of artefacts.

## CONCLUSION

In summary, both acute single exercise and combined exercise can significantly improve the performance of the executive task. However, acute combined exercise has a general facilitative effect on the performance of the naming task. Combined exercise was shown to be superior to single exercise for cognitive function, which was supported by task-efficient cerebral oxygenation and improved oxygen utilization during cortical activation in older individuals. In addition, in order to maximize cognitive function it may be important for older adults to take part in more cognitive demand exercise or take more kinds of exercise.

### Funding

The authors received no funding for this work.

### Competing Interests

The authors declare there are no competing interests.

### Author Contributions

- Zhiguang Ji conceived and designed the experiments, performed the experiments, analyzed the data, contributed reagents/materials/analysis tools, prepared figures and/or tables, authored or reviewed drafts of the paper, approved the final draft.
- Tian Feng and Lingnan Mei performed the experiments.
- Anmin Li conceived and designed the experiments.
- Chunhua Zhang contributed reagents/materials/analysis tools.

### Human Ethics

The following information was supplied relating to ethical approvals (i.e., approving body and any reference numbers):

The study was conducted ethically and received approval from the Ethics Committee of Shanghai University of Sport (No. 2017032).

### Data Availability

The raw measurements are available in Dataset S1. The raw data shows all results which show changes of oxygenated and reacton time.

### Supplemental Information

Supplemental information for this article can be found online at http://dx.doi.org/10.7717/peerj.7418#supplemental-information.

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
