# Peer review of "Influence of acute combined physical and cognitive exercise on cognitive function: an NIRS study"

_PeerJ, doi:10.7717/peerj.7418_

## Round 0.1 · original submission · Minor Revisions

The manuscript has not yet been sent out for peer review as I noticed the statistical results are incompletely reported.

All statistical results should be fully reported, including the the exact p-value. Error bars should be added to Figure 2. It is also recommended to overlay bar graphs with scatter plots showing individual data points, see e.g. https://doi.org/10.1371/journal.pbio.1002128.

Once these changes have been made, I'll send the manuscript out for peer review.

---

## Round 0.2 · Major Revisions

Although the reviewers thought the study was important, they raised several critical concerns. All three reviewers commented on language errors that made the manuscript confusing at times. Detailed feedback is provided to address these. Also questions were raised about the figures and interpretation of the results. I'd like to give the authors the opportunity to address these questions and rewrite the manuscript. Once these issues have been clarified, the manuscript will be reviewed to determine whether it meets the criteria for publication.

Reviewer 1 ·

Basic reporting

1. The English language of the article should be improved. Current English is hard to understand and not clear. Also, some words (like influence and join) were not selected properly and it causes reading difficulties.
2. Both introduction and discussion parts have some problems related with the references. Please cite the necessary articles for the fNIRS and NIRS systems (line 75-79) and check the bibliography for double cited references (Ludyga et al).
3. There are many missing literatures in relation with the cognition and exercise such as Byun et al 2014, Lee et al 2016, Browne et al 2016, Netz et al 2016, and Peruyero 2017 (http://dx.doi.org/10.1016/j.neuroimage.2014.04.067, https://doi.org/10.1186/s13063-016-1650-4, http://dx.doi.org/10.1016/j.rppede.2016.01.005, 10.1016/j.bandc.2016.08.002, https://doi.org/10.3389/fpsyg.2017.00921). Please consider these articles when you are describing the gap in the literature.
4. The authors should clearly describe the motivation behind the selection of older adults to the study and link that motivation with the hypothesis. In other words, why did the authors investigate the inhibition control with the older adults? Please justify your aim accordingly and re-write the introduction with that motivation.

Experimental design

1. The main research question is missing. Age of the participants should be noted.
2. The authors should explain the cognitive task section better. The readers want to know more detail without searching cited articles. A figure that describes the experimental procedure would be helpful. In current construction of the paper, it is hard to replicate the study.
3. The authors stated that there are four conditions with randomized order in current experimental design. But, this in not clearly stated on the manuscript. From text, someone could easily understand that only three of them counterbalanced across the participants. Please explain it better.
4. Please indicate the mean age on the text and describe how did you reach the participants.
5. The authors stated that they used 20 channel fNIRS system but in that section only 14 of them were used. Please describe why you did not use all fNIRS channels? Also, the authors cited figure 1 in here, but figure 1 does not contain any fNIRS related elements. Please correct it and provide a figure, which shows the connected fNIRS probe in a participant or head model.
6. Please describe the fNIRS analysis section better. For example, your sampling rate is 3.91 Hz and the duration of single stroop task is around 1 min. Please justify how did you get 20 values for each condition in your raw data? Did you average the oxyhemoglobin levels during single task?

Validity of the findings

1. In conclusion, the authors stated that “acute single exercise, single cognition and combined exercise enhanced the performance of the executive task”. How did the authors state this? It should be considered that there is slightly enhancement in the control condition as well. Please change that statement for the significant results or describe that better. Additionally, check the English on that section.
2. The results section have some uncompleted sentences and problematic figures. The legends of figure 2 should be revised according to the text. Also, you provide figure 2a and 2b but there is only one legend for them, which is not descriptive. Please correct them. Also, the figure 2a contains same graph headings. Please correct them.
3. The results are clear. However, the obtained results are not well discussed with the related literature in the discussion part. Especially, missing articles in parallel with current study is reducing the quality of this manuscript. Additionally, some of the literatures given in introduction is not discussed in the manuscript. Please re-write the discussion part accordingly.
4. Another important point is the missing discussion of the current results in parallel with age. Please re-shape the discussion and include the age discussion to this section.
5. Some of terms are really confusing in figures such as l-DLPFC, r-DLPFC. I suggest to use L-DLPFC and R-DLPFC instead of l-DLPFC, r-DLPFC.
6. There are some missing statements such as pro-test. Please control the grammar and word mistakes.

Additional comments

1. The aim of this study is to investigate the effects of different physical and cognitive load on inhibition control and cerebral oxygenation. This study is trying to establish the relationship between inhibition control and physical/cognitive loads. Additionally, PFC lateralization during the inhibition process is another topic in this manuscript. However, the manuscript has some serious problems in regard to figures and language errors (such as wrong words, incomplete sentences etc.). Also, some of the missing/not considered literatures are critically important for this study. The authors should re-write the introduction and discussion part accordingly.

2. In bibliography, the abbreviation of the journal in reference number 2 is correct. But, the PeerJ guidelines clearly indicated that the authors should write the full title of Journal. Please double check your bibliography according to that statement.

Reviewer 2 ·

Basic reporting

The paper comes across as well considered and provides a nice overview of the relevant literature. There are sections where the sentence structure makes it hard to follow (especially in the discussion), which could benefit from English language editing. I have highlighted a few examples under general comments.

The use of abbreviations in the results section hampers readability; the C, CE, SC, SE conditions could easily be replaced by control condition, cognitive exercise, physical exercise, and cognitive+physical exercise.

The titles in Figure 2 seem incorrect (all plots are titled “L-VLPFC”, this needs to be corrected) and the bar subplots in Figures 2 and 3 do not seem to add anything to the more detailed subplots. I would suggest to remove these bar subplots, and add the symbols indicating significant differences to the remaining subplots.

Experimental design

Based on the introduction, the paper seems to provide novel insights. The research question is well defined and clear. The methodology seems rigorous, although I cannot comment on technical specifics. It is not entirely clear whether a baseline fNIRS measurement is repeated before every Stroop task condition; I can imagine that increased skin blood flow due to physical exercise could affect the fNIRS substantially. Please clarify if this was the case.

The paper does not mention ethical approval; a statement that ethical approval for the study was obtained from an appropriate committee would need to be added.

Validity of the findings

The study investigates oxygenated blood flow to the prefrontal cortex (using fNIRS) as a potential explanation for changes in cognitive function (mainly inhibitory function) after cognitive, physical or combined cognitive and physical exercise. A reason for recruiting (young) older people is not provided but could be added if relevant.

Additional comments

Major:
1. It is unclear whether ethical approval was obtained for this research.
2. It is unclear whether Oxy-hb is expressed as change compared to a baseline measurement right before the conditions of interest (to minimize increased blood flow artifacts). If not, this could drive the findings with respect to increased Oxy-hb after physical exercise.
3. The English language should be improved to ensure that an international audience can clearly understand the text. Examples are: L78, L85-86, L145, L156, L254, L260, L261-267, L285 – the current phrasing makes comprehension difficult.

Minor:
• L71, “In recent studies assessed by …”: should this be “In recent studies that performed human brain imaging, a change in brain oxygenation it has been … intervention ...”.
• L80: how does the higher activation related to function? Is more better or is there a saturation effect?
• L82, “prefrontal”: should this be “prefrontal areas”?
• L93, “demand of exercise”: “may be one of the potential …”
• L95, “the question whether in combined exercise exceeds single exercise in acute exercise…”: please revise to clarify that this refers to a combination of (I assume) cognitive and physical exercise.
• L98, “Dual-task …”: the link of this sentence to the rest of the section is unclear, consider revising.
• L102, “According to neurovascular coupling …”: should this be “neurovascular coupling theory”? Plus: does the metabolic activity in cerebral blood flow increase or is the metabolic activity assessed as an increase in cerebral blood flow? Please revise this sentence for clarity.
• L105, “the first aim of the study is the oxygenation …”: is it to “effect of exercise on the oxygenation …”?
• L108, “while may be regulated”: please clarify what may be regulated.
• L113, earlier research: please specify the “different benefits” of Tai Chi and brisk walking on cognition.
• L116: Please specify what Ludyga et al. found and detail its relevance to the current study.
• L118, “…to address the gap which the effects …”: would this mean “to address the gap in knowledge on the effects …”?
• L134: please briefly explain the PAR-Q here with appropriate referencing.
• L137, “and cognitive function MMSE.”: “and cognitive function was assessed using the MMSE”.
• L137, “… a reading control condition was complicated …”: “was completed”. Please add a rationale for this control condition.
• L139: why was the control condition not randomized with the other 3 conditions? Could this have affected the results?
• L143: Please clarify what VFT means.
• L147: please add the rationale for allowing people to practice before the assessments. Please also clarify when the control condition was assessed.
• L150: how was the fNIRS position standardized?
• L157: which two heart rate values?
• L161: please clarify the naming and executive tasks. Where these a simple color recognition task and the incongruent condition of the Stroop test?
• L166: would this method of responding make the task even more difficult (since there is now a working memory component (recall of mapping) as well)? How would this affect the interpretation of the results? Would a difference score of “executive condition” minus “naming condition” provide a more pure measure of inhibition?
• L167: were the cues always presented as the letters “XXX” displayed in different font colors? I am unfamiliar with this task, what was the rationale for not using the congruent condition of the Stroop test?
• L169, “… and part of the stimuli is a rectangle appeared around the word, …”: I am not sure what this means, please clarify in the manuscript.
• L187: please provide the appropriate references to the software including version numbers.
• L190: would analysis of the error trials provide additional insight?
• L197: please clarify that the 4 x 2 RM ANOVA was 4 (conditions) x 2 (assessments). Consider to give the “naming” and “executive” task a more intuitive name; I’d consider both to require executive functions (especially since the response needs to be provided by pressing a keyboard key).
• L222-225: please clarify that the conditions were significantly higher “in oxy-hb levels”.
• L222: Should the posthoc tests be corrected for multiple comparisons?
• L226: “… there was a significant main effect …”.
• L239: please discuss the specific findings here: which conditions had a higher oxy-hb level, and which conditions did not differ in oxy-hb.
• L239, “… regardless of combined cognition”: “regardless of the addition of cognitive exercise”.
• L242: Please discuss potential explanations of this finding.
• L248 (and more often), “protest and posttest”: “pretest and posttest”.
• L252: please specify what Ludya et al found and why it is relevant to this study.
• L254: are these sensitivity results? Please clarify what was done, how and why.
• L259: please provide a reference for this finding. I assume this might be Hsieh et al. 2016, which is mentioned in the next sentence, but this should be made clear in this sentence as well.
• L274, “… reporting bilateral LPFC”: “LPFC activity”.
• L284: Would this be “A potential explanation for this finding”? Plus: the reference style seems off here.
• L285, “explicated”: “showed”?
• L293, “that brings stimuli’s information”: “involved in processing/recognition of visual stimuli”?
• L307, “only acute combined exercise”: both SE and CE show an effect, please revise.
4. Discussion: The results indicate that both CE and SE improve performance on the cognitive tests and increase PFC oxygenation. However, the paper does not discuss whether the additional cognitive component in CE amplified the effects of acute physical exercise on cognitive performance while this would be of interest to readers. Consider to add discussion of CE vs SE.
• Figure 1: What do the error bars depict, are these SD or SEM?
• Figures 2 and 3: Please add a line depicting the average effect to the subplots.
• Table: please correct the precision of BMI and VO2peak, 2 decimal digit seems sufficient. Should “MMSN” be “MMSE”? Please discuss the relevance of the VO2peak here and consider to add the heart rate measurements to this table.

·

Basic reporting

The standard of English was very poor, with grammatical errors, inappropriate use of words and many repetitions. In places it was difficult to understand what information was being conveyed.

The concept of 'inhibition control' which the study focuses on was not well defined.
A topic very relevant to the study that was not discussed adequately or referenced sufficiently is vascular regulation of the cortex and how exercise may change this long term.

'Acute' exercise needs to be defined.

Inadequate references are given. For example, in line 337, the MMSE was selected without appropriate referencing.

Experimental design

The research fits the Aims and Scope of the journal.

It is not clear how the fNIRS analysis was performed. In lines 191 - 192, it is stated that the change is calculated by 'subtracting the mean Oxy-Hb of the pre-intervention task period from the mean Oxy-Hb of the post-intervention period'. However in lines 182 - 184 it discusses changes in concentration compared to baseline.

How was the position of the optodes relative to the cortical region (i.e. DLPFC, VLPFC) assessed? Were MRIs taken or a 3D digitiser used?

Description of the naming and executive condition used in the Stroop test is needed.

How long after the acute exercise was the fNIRS measured?

Were any Bonferroni (or other) adjustments made for the multiple comparisons?

Validity of the findings

The standard deviations in figure 1B for the executive condition are very high. The significant main effect of time - p<0.001 is surprisingly. Were the data normally distributed and were there outliers which could account for the significant change reported?

---

## Round 0.3 · Minor Revisions

The revised manuscript has improved considerably but the reviewers have several additional comments that need to be addressed before the manuscript can be accepted for publication. In particular, potential artefacts in fNIRS recordings should be properly addressed.

Reviewer 1 ·

Basic reporting

The improvement of the MS is very satisfactory now.

Experimental design

I still have some questions which were placed in general comments.

Validity of the findings

No comment

Additional comments

Thank you for your efforts to enhance the quality of the paper. Nearly, all of the suggestions from my side were completed accordingly. However, I have still further minor revisions for the paper.

Abstract
Line 58: The fNIRS findings show that acute single exercise influences oxygenation for executive tasks but not oxygenation for naming tasks. Should be replaced as “The fNIRS findings showed that acute single exercise influences oxygenation for executive tasks but not for naming tasks.

Line 59: Greater increment was observed in the post-exercise session of combined exercise during the modified Stroop. Replaced with the “Greater improvement was observed in the post-exercise session of combined exercise during the modified Stroop.”

Introduction
Line 90: “improved” should be replaced with “improve”

Line 98: “Moreover, few studies studied the mode of acute exercise.” should be replaced with “Moreover, there are limited number of studies in which the effects of different types of exercise were investigated”

Methods
Line 177: VFT abbreviation is used first time in this section besides the abstracts. So, please add the “Verbal Fluency Task” before the VFT abbreviation.

Line 181-182: “The cognitive + physical exercise (CE + PE) paradigm combined walking at the same intensity as single exercise with cognitive + physical exercise.” This sentence is not clear. What do you mean with walking at the same intensity?

Line 209: Experimental procedure figure for whole paradigm would be still helpful. There is no explanation about the stages for combined exercise. When did you apply the cognitive tasks during the CE+PE condition? Please describe it better.

Line 229: The figure of the optodes were not good enough for the demonstration in regard to resolution. Please consider the literature you already referred in the manuscript. At least 5 or 6 of them represented the fNIRS optodes in a better way. You can easily take a photograph of a participant forehead and mark the ROI on that.

Line 233: “filtered” should be replaced with the “filter”

Line 243: Please correct the term as SPSS (v19.0).

Results
The resolutions of the figure 3 and 4 should be noticed and improved. Especially, figure 4 is hard to read due to this problem.
Line 249 and 261: Please revise the first sentence of the response time and accuracy related results section. For example: “When the response time was considered, in the naming task,…”
Line 257: “pro” will be “pre”

Discussion
Line 286: “modes of exercise” can be replaced with the “type of exercise”

Reviewer 2 ·

Basic reporting

The English has been improved considerably. There are a few sentences that could be improved, and I hope that my co-reviewers point these out, but overall the paper is understandable to a general scientific audience.

The new Figure 1 is a great addition. The current figures 2 to 4 are of very low quality in the reviewer's version, please provide a high-resolution version with the revision.

Experimental design

I am still not convinced that the findings are not influenced by an artefact in fNIRS due to increased skin blood flow after physical exercise. It was unclear from the author's response if they considered or accounted for this. If they haven't, this should be discussed as a limitation to this study. If they have, this should be made more clear in the article.

Validity of the findings

No comments

·

Basic reporting

Improved language structure.
More information on changes in cognitive function in ageing would be appropriate.
Figure 3 needs editing.
The aims need to be more clearly presented.

Experimental design

Pleas describe how the optodes were placed.

Validity of the findings

Limitations of fNIRS analysis should be added with discussion of artefacts related to talking and changes in haemodynamics on the different days.

Additional comments

General comments

The manuscript is much improved and many points clarified.
However, there are still grammatical errors eg. Lines 167-171.

All acronyms need to be defined eg VLPFC, DLPFC, LPFC

Methodology is not clearly described. For example, how long after the exercise conditions was the modified Stroop condition applied?
The fNIRS was recorded at rest before the intervention and again for one minute after the intervention. Was this during the modified Stroop test when the participant was talking, which would have introduced an artefact in the signal?

The naming condition needs to be better defined and more description added to Figure 1.

In response to the question regarding placement of electrodes the reply was ;’…the software integrates a 3D digitiser…’. Please add this to the text with reference.

Again, standard deviations are very high in Figure 3B and yet p < 0.001. Please provide a scatter plot of the individual data.

In figure 3 change the ‘C’ along the x-axis to ‘RC’. Change ‘SC’ to ‘CE’, ‘SE’ to ‘PE’ and ‘CE’ to ‘CE&PE’.

Table 3 – typos ‘Hight’, ‘MMSN’. Change 0.45 in M/F to number of male/female participants.

---

## Round 0.4 · Major Revisions

I did not send out the revised manuscript for peer review, as this is already the third version. When reviewing the revised manuscript myself, I noticed that the authors did not fully address the reviewer comments and that the manuscript still contains many errors. As such, the revised manuscript is still not acceptable for publication in PeerJ. I would like to give the authors a final chance to address these issues by making the following changes.

General comments

It remains unclear in the manuscript when fNIRS was acquired. Please specify in the Methods section during which conditions fNIRS was assessed. Also, clarify which conditions were part of the intervention and which were used to test changes in cognitive function.

In their rebuttal the authors stated that participants didn’t talk during the modified Stroop test. This is not clear from the description in the Method section (l. 204-212): “In the naming task, a visual stimulus was presented as ink colors of XXX, the participants were instructed to read the colour of the ink.” To avoid confusion, I would suggest to rewrite: “In the naming task, a visual stimulus was presented as ink colors of XXX and participants were instructed to indicate the color of the ink by pressing a response key (i.e., red—J key, blue—F key).” And at the end of the paragraph “Participants did not speak during either the naming or the executive task to avoid artefacts resulting from speaking.”

The authors should discuss potential artefacts of fNIRS recordings in the limitation section of the Discussion (l. 363-370). What kind of artefacts can be expected following physical exercise (or during speaking) and what did they authors do to minimize these artefacts?

Many figure panels are not described in the figure legend. For example, figure 3 has 6 panels (A-F), but only panels A and B are described in the figure legend. In figure 4 none of the panels are described. Please describe all panels in the figure legends.

Please reduce the number of abbreviations that is used in the text, as it is confusing for the readers and there is no page limit at PeerJ.


Specific comments

- Abstract: change “immediately after the experimental condition” as fNIRS was only assessed after heart rate returned to baseline levels
- Abstract: rephrase “but also the different results between combined exercise and single exercise”. It is unclear what is meant and the sentence is grammatically incorrect.
- Line 150: replace “Participants (0.45 in f/m” with “Twenty participants (9 female”
- Line 164: Write out “MMSE” in full
- Line 166: replace “On laboratory visits 1, 2, 3, and 4” with “On laboratory visits 1-4,”
- Line 172: replace “heart rate returns” with “heart rate returned”
- Line 176: please define “SC”. Or should this be “RC”?
- Line 190: replace “(HR), as assessed by” with “(HR) was assessed using”
- Line 202: replace “centre” with “center”
- Line 206: replace “colour” with “color”
- Line 210: replace “colour” with “color”
- Line 221: remove “For further confirmation,”
- Line 223: replace “3 dimensional” with “3-dimensional”
- Line 226: replace “(1–4 channels)” with “(channels 1–4)”. Same for other channels
- Line 227: replace “(5、6、7、11 channels)” with “(channels 5, 6, 7 and 11)”
- Line 241: replace “errors” with “error”
- Line 249: replace “Behavioral and Oxy-Hb data were analyzed using SPSS (v19.0 ) (conditions) × 2 (assessments) repeated measure” with “Behavioral and Oxy-Hb data were analyzed in SPSS (v19.0 ) using a 3 (conditions) × 2 (assessments) repeated-measures ANOVA.”
- Line 251: replace “Post hoc” with “post-hoc”
- Line 252: replace “..” with “.”
- Figure 4A and 4F: are the labels “C, SC, SE, CE” correct? Or should this be “RC, CE, PE, PE+CE”?
- Line 271: Please rephrase “, but also the different results between combined exercise and single exercise.” It is unclear what is meant and the sentence is grammatically incorrect.
- Line 330: replace “control and cognitive condition” with “the control and cognitive condition”
- Line 331: replace “to previous studies that” with “to previous studies which showed that”
- Line 334: replace “VLPFC, and DLPFC” with “VLPFC and DLPFC”

---

## Round 0.5 · Minor Revisions

Not all of the previous comments were addressed and some of the revisions introduced new typos. Once these final changes are made the manuscript can be accepted for publication.

Abstract: replace “immediately after the experimental condition” with “after the experimental condition when the heartrate returned to baseline”

Line 157: replace “when the heart rate returns” with “when the heart rate returned”

Line 207: replace “channels 5、6、7、11” with “channels 5, 6, 7 and 11”

Line 338-340: replace “Meanwhile artefacts of speaking may have an impact on the results. Although the language area is not ours ROI. The further studies can optimize modules to minimize impact.” with “Artefacts due to speaking may have impacted the results, but the language area was not our ROI. Further studies could optimize task modules to minimize the impact of artefacts.”

Replace figure legend 3 with: “Figure 3, Reaction time of the naming task and executive task in the modified Stroop task: RC, Reading control condition; CE, cognitive exercise; PE, Physical exercise; CE+PE, cognitive + physical exercise. Reaction time of the naming task (A) and executive task (B); (C) scatter plot of the pre-test during naming task; (D) scatter plot of the pre-test during executive task; (E) scatter plot of the post-test during naming task; (F) scatter plot that the post-test during executive task. *Significant difference between pro-test and post-test, p < 0.05.”

Replace figure legend 4 with: “Figure 4, The interaction effects of the naming task and executive task on task-related changes in Oxy-Hb levels: RC, Reading control condition; CE, cognitive exercise; PE, Physical exercise; CE+PE, cognitive + physical exercise. Oxy-Hb levels of the naming task (A) and executive task (F); (B) scatter plot of the Oxy-Hb levels of l-DLPFC during the naming task; (C) scatter plot of the Oxy-Hb levels of l-VLPF during naming task; (D) scatter plot of the Oxy-Hb levels of r-VLPF during the naming task; (G) scatter plot of the Oxy-Hb levels of r-DLPF during the executive task; (H) scatter plot of the Oxy-Hb levels of l-DLPFC during executive task; (I) scatter plot of the Oxy-Hb levels of l-VLPF during executive task; (J) scatter plot of the Oxy-Hb levels of r-DLPF during executive task.*Significant difference between four conditions, p <0 .05.”

---

## Round 0.6 · accepted · Accept

The authors addressed the outstanding comments